# An Advanced in Silico Modelling of the Interaction between FSCPX, an Irreversible A_1_ Adenosine Receptor Antagonist, and NBTI, a Nucleoside Transport Inhibitor, in the Guinea Pig Atrium

**DOI:** 10.3390/molecules24122207

**Published:** 2019-06-12

**Authors:** Adrienn Monika Szabo, Tamas Erdei, Gabor Viczjan, Rita Kiss, Judit Zsuga, Csaba Papp, Akos Pinter, Bela Juhasz, Zoltan Szilvassy, Rudolf Gesztelyi

**Affiliations:** 1Department of Internal Medicine, Faculty of Medicine, University of Debrecen, H-4032 Debrecen, Hungary; szabo.adrienn23@gmail.com; 2Department of Pharmacology and Pharmacotherapy, Faculty of Medicine, University of Debrecen, H-4032 Debrecen, Hungary; erdei.tamas@pharm.unideb.hu (T.E.); vicgabaa@gmail.com (G.V.); kiss.rita@med.unideb.hu (R.K.); juhasz.bela@med.unideb.hu (B.J.); szilvassy.zoltan@med.unideb.hu (Z.S.); 3Department of Health Systems Management and Quality Management for Health Care, Faculty of Public Health, University of Debrecen, H-4032 Debrecen, Hungary; zsuga.judit@med.unideb.hu (J.Z.); dr.papp.csaba@gmail.com (C.P.); 4Institute of Mathematics, Faculty of Science and Technology, University of Debrecen, H-4032 Debrecen, Hungary; apinter@science.unideb.hu

**Keywords:** adenosine, CPA, FSCPX, NBTI, A_1_ adenosine receptor, operational model of agonism, receptorial responsiveness method, RRM, computer simulation

## Abstract

In earlier studies, we generated concentration-response (E/c) curves with CPA (*N*^6^-cyclopentyladenosine; a selective A_1_ adenosine receptor agonist) or adenosine, in the presence or absence of *S*-(2-hydroxy-5-nitrobenzyl)-6-thioinosine (NBTI, a selective nucleoside transport inhibitor), and with or without a pretreatment with 8-cyclopentyl-*N*^3^-[3-(4-(fluorosulfonyl)-benzoyloxy)propyl]-*N*^1^-propylxanthine (FSCPX, a chemical known as a selective, irreversible A_1_ adenosine receptor antagonist), in isolated, paced guinea pig left atria. Meanwhile, we observed a paradoxical phenomenon, i.e., the co-treatment with FSCPX and NBTI appeared to enhance the direct negative inotropic response to adenosine. In the present in silico study, we aimed to reproduce eight of these E/c curves. Four models (and two additional variants of the last model) were constructed, each one representing a set of assumptions, in order to find the model exhibiting the best fit to the ex vivo data, and to gain insight into the paradoxical phenomenon in question. We have obtained in silico evidence for an interference between effects of FSCPX and NBTI upon our ex vivo experimental setting. Regarding the mechanism of this interference, in silico evidence has been gained for the assumption that FSCPX inhibits the effect of NBTI on the level of endogenous (but not exogenous) adenosine. As an explanation, it may be hypothesized that FSCPX inhibits an enzyme participating in the interstitial adenosine formation. In addition, our results suggest that NBTI does not stop the inward adenosine flux in the guinea pig atrium completely.

## 1. Introduction

The A_1_ adenosine receptor, a member of the adenosine receptor family (formerly known as P_1_ purinoceptors), exerts extensive regulatory (mainly protective and regenerative) functions in almost all tissues [1,2], including the myocardium [3]. As a protective action, the A_1_ adenosine receptor mediates strong negative inotropic effect consisting of an indirect component (decreasing the stimulated contractile force, seen in both the atrium and ventricle) and a direct one (reducing the resting contractile force, only characteristic of the atrium in most species) [4].

In earlier ex vivo studies [5,6] carried out in isolated, paced guinea pig left atria (a simple and reliable model to investigate the myocardial adenosinergic system), we observed a paradoxical phenomenon concerning FSCPX, a chemical widely known and used as a selective, irreversible A_1_ adenosine receptor antagonist [7,8,9,10]. Namely, in the presence of NBTI, a selective and potent inhibitor of the nucleoside transporter type ENT1 (the main carrier for the myocardial adenosine transport) [11,12], FSCPX pretreatment appeared to enhance the maximal response to adenosine, the physiological full agonist for the A_1_ adenosine receptor. Back then, we considered this phenomenon as a misleading plotting peculiarity that was caused by neglecting the effect evoked by the surplus endogenous adenosine accumulated due to NBTI in the cardiac interstitium [5].

In a subsequent study [13], we in silico reconstructed some concentration-response (E/c) curves selected from [6]. Based on the behavior of the simulated E/c curves of different adenosine receptor agonists, we have hypothesized that pretreatment with FSCPX alters the influence of NBTI on the E/c curves. As a mechanism, we have assumed that FSCPX may modify ENT1 (the equilibrative and NBTI-sensitive nucleoside transporter [11,12]) in a way that ENT1 preserves its ability to transport adenosine but NBTI can less inhibit this transport [13].

Next, we tested this putative effect of FSCPX in the isolated, paced guinea pig left atrium [14]. Based on results of that study (Figure 1), we have propounded a new hypothesis, i.e., FSCPX pretreatment inhibits only one effect of NBTI on the E/c curves of adenosine receptor agonists, the one that is mediated via increasing the interstitial concentration of endogenous adenosine. The other effect of NBTI is mediated by elevating the interstitial level of exogenous adenosine (if any), and that action is proposed to remain intact after an FSCPX pretreatment. As a mechanism for this phenomenon, we have supposed that FSCPX may inhibit one (or some) enzyme(s) participating in the interstitial formation of adenosine [14], an action not acknowledged thus far.

Addressing the strict distinction between endogenous and exogenous adenosine cannot be overemphasized given that in our experimental conditions, elevation in the interstitial level of endogenous versus exogenous adenosine exerts the opposite effect on the E/c curve of adenosine [5,6]. In general, NBTI, by blunting the normally inward transmembranous adenosine flux in the heart and thereby preventing adenosine from the intracellular elimination [16,17,18], increases the interstitial level of adenosine of both origins. However, endogenous adenosine is accumulated by NBTI **before** the generation of the E/c curve, thus it consumes (in part) the response capacity of the A_1_ receptors and thereby decreases the observable effect evoked later by an exogenous agonist (used for the E/c curve). In contrast, exogenous adenosine is accumulated by NBTI **during** the construction of the E/c curve, so it can elicit a greater effect. A distinction between effects of endogenous and exogenous agonists (in experimental arrangements such as the present one) forms the basis for the so-called receptorial responsiveness method (RRM) [19,20], theoretical concept of which was used for the current work too. It is also important that CPA, a synthetic A_1_ adenosine receptor full agonist, is relatively resistant to the adenosine-handling enzymes [15], so its level is minimally affected by NBTI.

The goal of the present study was to revisit the issue of the above-mentioned paradoxical phenomenon, and to in silico reevaluate the major conclusions of our relevant ex vivo [5,14] and in silico [13] investigations. For this purpose, eight (averaged) E/c curves, based on which these conclusions were drawn, were selected (Figure 1, Table 1) and reproduced in silico herein. Simulation was made using different assumptions, and then the different models were compared.

## 2. Results

### 2.1. Models 1 and 2

In Models 1 and 2 (which differ from each other in their E_m_ parameter, 100 and 99, respectively), the simplest conditions were assumed to simulate the guinea pig atrial adenosine homeostasis and A_1_ adenosinergic control of contractility. A complete blockade of the transport of agonist A (representing adenosine) in response of agent NB (representing NBTI) was supposed with no interaction between effects of agent X (symbolizing FSCPX) and agent NB. While the simple E/c curves (i.e., curves without any agent NB treatment; see Section 4.3) could be simulated well, the complex E/c curves (receiving an NB treatment; see Section 4.4) considerably differed from the original ex vivo E/c curves (seen in Figure 1), regarding their positions and, for the X + NB co-treated E/c curves, their shapes too (Figure 2). Thus, neither Model 1 nor Model 2 were suitable to properly model the biological system (represented by Figure 1 and Table 1).

The main conclusion drawn from these two early models is that E_m_, a parameter of the operational model defining the maximal signal amplification ability of the system, has a profound impact on the behavior of our complex E/c curves. Namely, if E_m_ was 100 (more precisely, if E_m_ was in an interval of 100–(~99.9), both solely NB-treated E/c curves exceeded the corresponding X + NB co-treated E/c curves (irrespectively of the exogenous agonist used). In turn, if E_m_ was below this value (in fact, below the above-mentioned value range), the X + NB co-treated E/c curves surpassed the corresponding NB-treated E/c curves (Figure 2).

At about E_m_ = 99.9, the corresponding complex E/c curves practically coincided (data not shown). This observation approximated the lower limit of the range for E_m_ (100–(~99.9)), in which the paradoxical phenomenon (i.e., FSCPX seemingly increases the response to adenosine) did not occur. It should be noted that E_m_ values higher than 100 do not have biological meaning in the ex vivo system to be simulated. Accordingly, raising E_m_ above 100 led to, first, enormously high, then negative E_max_ values for the complex E/c curves (data not shown).

### 2.2. Model 3

Model 3 differed from the earlier ones regarding three properties: (1) E_m_ was set to 90, (2) c_bias_ (see: Equation (2) in the Section 4.4) was set to a c_x_ value measured previously in the original ex vivo system (in the case of the sole NBTI treatment) [14], and (3) a slow agonist A transport was supposed under the effect of agent NB (instead of the total stop seen in Models 1 and 2, furthermore in Model 4-v1 below). Model 3 acceptably simulated all simple E/c curves and the solely NB-treated E/c curves. However, both X + NB co-treated E/c curves showed substantial deviation from the corresponding ex vivo E/c curves seen in Figure 1 (Figure 3). Thus, an interference should be assumed between effects of agents X and NB.

### 2.3. Model 4

Model 4 only differed from Model 3 in that it considered the effect of agent NB on the level of the endogenous agonist A with two distinct c_bias_ values (both obtained from [14]), separately for the pure NB treatment and the X + NB co-treatment (assuming an interaction between agents X and NB). The fact that c_bias_ for the X + NB co-treatment was smaller than that for the sole NB-treatment modelled that agent X inhibited the action of agent NB to increase the concentration of the endogenous (but not exogenous) agonist A. Contrary to the previous models, Model 4 adequately simulated all E/c curves (cf. Figure 3 and Table 2 with Figure 1 and Table 1, respectively). It is also important to note that moderate changes of values (in both directions) of the model parameters did not cause dramatic changes in the E/c curves (contrary to that was experienced in the case of E_m_ close to 100).

### 2.4. Models 4-v1 and 4-v2

Model 4-v1 and Model 4-v2, variants of Model 4, modified only the complex E/c curves of agonist A and the X + NB co-treated E/c curve of agonist A, respectively, in comparison with Model 4.

Model 4-v1 differed from Model 4 in assuming the complete inhibition of the agonist A transport by agent NB (again). The E/c curves of agonist A subjected to any agent NT treatment showed less similarity to the ex vivo counterparts than the corresponding curves in Model 4 did, regarding their position (cf. Figure 3 and Figure 4 with Figure 1). Consequently, in our in silico system, a weak agonist A transport should be assumed in the presence of agent NB, as shown by Model 4.

In turn, Model 4-v2 differed from Model 4 in supposing that a pretreatment with agent X inhibited both actions of agent NB (i.e., the increase of levels of both endogenous and exogenous agonist A). Accordingly, in Model 4-v2, in addition to the use of two c_bias_ values, a weakened inhibition by NB on the agonist A transport after the agent X pretreatment was considered as compared to the solely NB-treated state (modelling that agent X blunted the action of agent NB on the level of exogenous agonist A). This resulted in a dislocation of the X + NB co-treated E/c curve of agonist A as compared to the corresponding curve in Model 4 as well as in the original ex vivo system (cf. Figure 4 with Figure 1 and Figure 3). Consistent with this, any decrease in the extent of the supposed influence of agent X on the level of the exogenous agonist A ameliorated the position of the X + NB co-treated E/c curve of agonist A (data not shown). Thus, Model 4-v2 also yielded a worse outcome than Model 4.

Taking all together, Model 4 conferred the best similarity to the original ex vivo E/c curves (presented in Figure 1), so Model 4 has been regarded the final model of the present work.

## 3. Discussion

In the present in silico study, we have found that E_m_ parameter of the operational model of agonism [21] can influence the behavior of our complex E/c curves (reflecting the net of actions of two agonists while not accounting for one of them), i.e., the rank order of curves regarding the maximal response can be changed, an effect not characteristic of our simple E/c curves (expressing the action of one agonist in a standard way). Furthermore, we have gained in silico evidence for an interference between effects of FSCPX and NBTI in our ex vivo experimental setting used in several earlier studies [5,6,14]. This finding extends beyond the well-established A_1_ adenosine receptor antagonist property of FSCPX, indicating an inhibitory action exerted by FSCPX on the interstitial adenosine accumulation produced by NBTI, a selective and potent blocker of the nucleoside transporter type ENT1. Regarding the mechanism of this interference, in silico evidence has been obtained supporting that FSCPX only inhibits the interstitial accumulation of endogenous (but not exogenous) adenosine. As an additional result, we have found that NBTI seems not to completely inhibit the inward adenosine flux in the guinea pig atrium.

Cardiac functions may be modelled using various approaches, including in silico [22,23], ex vivo [24] and in vivo [25] works. The present investigation was designed to simulate the main adenosine-handling mechanisms and the A_1_ adenosinergic control of contractility in the guinea pig left atrial myocardium, similarly to our recent in silico study [13]. We have in silico reproduced eight E/c curves that were generated with CPA and adenosine in our recent ex vivo study [14]. During this procedure, four models (and two additional variants of the last one) were constructed, each one addressing a set of assumptions, in order to find the final model exhibiting the best fit to the ex vivo results (seen in Figure 1).

In light of the new findings, some conclusions of our previous in silico study [13] should be refined. In several earlier investigations carried out in our laboratory, a paradoxical appearance of the FSCPX + NBTI co-treated adenosine E/c curve relative to the FSCPX-naïve NBTI-treated adenosine E/c curve was observed, namely the irreversible antagonist FSCPX seemed to increase the maximal response of the adenosine E/c curve in the presence of NBTI, a nucleoside transport blocker [5,6,14]. In [13], this phenomenon could be in silico reproduced by supposing that FSCPX diminishes the effects of NBTI on the level of both endogenous and exogenous adenosine. Putting this earlier result into a biological context, this may indicate that FSCPX modifies ENT1 in a way that the altered ENT1 can carry adenosine well, but it can be less inhibited by NBTI. In the present work, however, the paradoxical phenomenon in question could be reproduced in an alternative way as well, namely by decreasing the E_m_ parameter of the operational model of agonism (as shown by Models 1 and 2 in Figure 2). Thus, the paradoxical appearance of the FSCPX + NBTI co-treated and only NBTI-treated adenosine E/c curves may indicate, but does not evidence, an interference between effects of FSCPX and NBTI. Nonetheless, an important conclusion of [13], i.e., it is necessary to construct an ex vivo FSCPX + NBTI co-treated CPA E/c curve in order to clarify whether the above-mentioned interference exists, proved to be useful. Owing to the improved experimental protocols used in [14], it has been demonstrated that FSCPX considerably reduces the interstitial accumulation of endogenous adenosine caused by NBTI, providing ex vivo evidence of the interaction between FSCPX and NBTI at the level of their effects.

In the operational model of agonism, a widely known quantitative model of receptor function, E_m_ represents the maximum of the transducer function (signal amplification) of the receptor system (consisting of a receptor together with its postreceptorial signaling), hence it indicates the achievable maximal effect in the given system [21,26]. In the present investigation, the unexpectedly great impact of whether E_m_ was set to 100 (or to a value very close to 100) or not may be associated with the fact that E_m_ = 100 is an absolute maximum for the present system. This is because effect values of our original ex vivo E/c curves (that were obtained from [14] and were simulated herein) had been defined as a percentage decrease of the initial contractile force, and the contractile force cannot be less than zero.

It should be noted that the operational model has been recently criticized for its some unfavorable features during curve fitting and for some conceptional deficiencies that make the interpretation of results problematic in some cases [26,27]. However, owing to its comprehensive and relatively flexible nature [21,26], and to the fact that it has an extended version capable of handling the co-action of two agonists in the same system [28], the operational model is suitable to serve as a framework for computer simulation where it is used to generate E/c curves.

Keeping the pivotal role of E_m_ in the simulation of our complex E/c curves in mind, further in silico models were evaluated: (1) Model 3 for the scenario that there is no interaction between FSCPX and NBTI, (2) Model 4 to represent that there is an interaction that affects only the fate of the endogenous agonist A (symbolizing adenosine produced in the myocardial interstitium), and (3) Model 4-v2 to represent the existence of such an interaction that affects the fate of both endogenous and exogenous agonist A (this latter one symbolizing the adenosine administered for the E/c curve). The relative position of the simulated X-treated and X + NB co-treated E/c curves of agonist C demonstrates that Model 4 and Model 4-v2 are more consistent with the ex vivo data than Model 3 (cf. Figure 1 and Figure 3), a finding that yields in silico evidence for an interaction between effects of FSCPX and NBTI. To gain insight into the underlying mechanism of this interaction, comparison of Model 4 with Model 4-v2 provided information. Regarding the relative position of the simulated NB-treated and X + NB co-treated E/c curves of agonist A, Model 4 shows a greater similarity to the original ex vivo system than Model 4-v2 does (cf. Figure 1, Figure 3 and Figure 4). Thus, it appears reasonable to conclude that FSCPX pretreatment blunts only the accumulation of endogenous adenosine caused by NBTI.

It is tempting to speculate about the exact molecular mechanism of FSCPX’s action other than irreversible antagonism on the A_1_ adenosine receptor. In our most recent ex vivo investigation, FSCPX has been assumed to blunt one (or some) enzyme(s) that participate(s) in the interstitial adenosine formation [14]. In this case, of course, FSCPX should inhibit the interstitial adenosine production in the absence of NBTI as well (i.e., upon intact nucleoside transport). An observation appears to contradict this conclusion, i.e., no clue of this inhibitory action is seen at the only FSCPX-treated E/c curves of adenosine receptor agonists (in comparison with the corresponding naïve E/c curves) (Figure 1). However, the lack of such a clue may be because the resting level of the endogenous adenosine in the interstitial fluid is too low to evoke a significant negative inotropic effect [29]. Therefore, a decrease in the resting interstitial adenosine concentration does not uncover itself in the E/c curves of adenosine receptor agonists. Thus, only maneuvers producing a significant increase in the interstitial concentration of endogenous adenosine, such as nucleoside transport blockade, can reveal this putative enzyme inhibitory action of FSCPX. To clarify this issue, further investigations are warranted. As a first step, performing inhibitor screening assays may be useful to explore whether FSCPX can blunt one (or some) ectonucleotidase(s), especially ecto-apyrase CD39 (a.k.a. lymphocyte surface protein CD39 or E-NTPDase1) and ecto-5′-nucleotidase (a.k.a. lymphocyte surface protein CD73), two important enzymes regarding the interstitial adenosine level in the heart [3].

An additional finding of the present investigation is that ENT1 blockade elicited by NBTI appears not to completely stop the inward transmembranous adenosine transport in the guinea pig atrium (as indicated by the comparison of Model 4 with Model 4-v1; cf. Figure 3 and Figure 4). It should be noted that Model 4 and Model 4-v1 contain an assumption based on the appearance of CPA and adenosine E/c curves generated in our ex vivo experimental setting. Namely, the only difference between CPA and adenosine, two full agonists for the A_1_ adenosine receptor, is the absence and presence of an inward transport, respectively. This assumption is supported by the fact that the synthetic CPA is relatively unaffected by adenosine-handling enzymes located mainly intracellularly, thus it is not transported in a significant manner, contrary to adenosine, a metabolite quickly transported into the cells [15]. This finding can be well explained with the existence of adenosine carriers other than ENT1 in the heart [30].

## 4. Materials and Methods

### 4.1. Properties of the Biological System to Be Simulated

The physiological adenosine metabolism in the heart implies a net interstitial production and net intracellular elimination resulting in a continuous inward transmembranous adenosine flux, implemented mostly (but not exclusively) via ENT1 [11,12,16,17,18,30]. Therefore, adenosine has shorter half-life than adenosine analogues more resistant to adenosine-handling enzymes, e.g. CPA [15].

Inhibition of ENT1 by NBTI prevents the intracellular elimination of interstitial adenosine, hence it elevates the interstitial level of endogenous adenosine and slows the decrease of the interstitial level of exogenous adenosine [18], without significantly affecting the interstitial CPA concentration [19]. Hence, NBTI influences the E/c curves of CPA and, especially, adenosine in a complex manner [14].

Based on its well-known irreversible antagonist property on the A_1_ adenosine receptor [7,8,9,10], the effect of FSCPX can be well explained in the atrial myocardium, if administered alone. Because of the great A_1_ adenosine receptor reserve related to the direct negative inotropic effect, FSCPX produces a dextral displacement of the E/c curve of both adenosine and CPA [10]. However, if it is used together with NBTI, the effect of FSCPX is no longer easy to interpret [5,6,14].

### 4.2. Simulation of the Adenosine Homeostasis and A_1_ Adenosinergic Control of Contractility in the Guinea Pig Atrial Myocardium

Broadly, the simulation was performed as it was described in our recent computer simulation study [13]. Some details were modified, as described in this subsection. To generate E/c curves simulating the effect of A_1_ adenosine receptor agonists on the contractile force of the isolated, paced guinea pig left atrium, the operational model of agonism was applied, both for the action of one agonist (Equation (1)) [21] and the co-action of two agonists (Equation (2)) [28]. The operational model provides a general, (fully) quantitative description of the relationship between the concentration of bioactive agents and the effect mediated by a receptor specific for the given agents. Moreover, this model contains the appropriate parameters, by means of which the effects of FSCPX (concentration of the operable receptors) and NBTI (parameters for two different agonists) can be considered [21,26,28].

To address the different impact of adenosine of endogenous and exogenous origins, a procedure, developed from RRM [19,20] and first described in [13], was applied (see: Equation (3) below). By means of this procedure, the neglect of one agonist from two co-acting agonists was simulated. The overlooked agonist concentration modelled the extra interstitial concentration of endogenous adenosine accumulated by NBTI, which came into being before the construction of an E/c curve with (exogenous) adenosine or CPA (agonists for the same receptor).

Using different input data (Table 3) and assumptions, four models and two additional model variants of Model 4 were defined that resulted in six sets of E/c curves, each set containing eight curves, four ones belonging to agonists C and four ones belonging to agonists A (Figure 2, Figure 3 and Figure 4). To characterize and illustrate the simulated E/c curves, the Hill equation (Equation (4)) was fitted to them.

Adenosine, the physiological agonist for the A_1_ adenosine receptor, the major adenosine receptor type of the supraventricular myocardium [3,4], was modelled with an agonist A. Based on its location, two agonist A concentrations were considered, one “in the organ bath” (a “bathing medium concentration”), and another one “at the receptors” (a “near-receptor” concentration). Based on its origin, an “exogenous” (administered to generate an E/c curve) and an “endogenous” (produced in the atrial tissue) agonist A were distinguished.

The inward transport of adenosine was simulated differently for the exogenous and endogenous agonist A. In the absence of a transport inhibitor, the concentration “in the organ bath” designated for the exogenous agonist A was divided by 400, when computing its effect. This maneuver simulated the fact that, in vivo or ex vivo, the concentration of an intensively transported agonist is lower at its receptors (in the interstitial fluid) than in the blood plasma or bathing medium. In the presence of a transport inhibitor, the concentration of the exogenous agonist A in the organ bath was not divided, or it was but by a number much less than 400 (6 or 14.8952), during the calculation of its effect. In turn, the surplus concentration of the endogenous adenosine, accumulated by a transport inhibitor, was considered as a c_bias_ value of agonist A, using arbitrary values or values measured in the most recent ex vivo study [14], when calculating its effect (Table 3). All E/c curves generated with the consideration of c_bias_ (i.e., all E/c curves reflecting the effect of a transport inhibitor) were regarded as “biased”. This is because it was simulated that c_bias_ and its effect were neglected during the evaluation of the raw E/c data. (Indeed, this is the case during a conventional evaluation of E/c data measured in the presence of a transport inhibitor that accumulates an unknown amount of the endogenous agonist for the given receptor.)

CPA, a synthetic agonist of the A_1_ adenosine receptor, was modelled with an agonist C. As CPA is eliminated by adenosine-handling enzymes to a much lesser extent than adenosine, the “bathing medium concentration” and “near-receptor concentration” of agonist C were considered to be equal. Accordingly, when calculated its effect, the “bathing medium concentration” of agonist C was never divided throughout the simulation.

FSCPX, an irreversible A_1_ adenosine receptor antagonist, was modelled with an agent X. The effect of a pretreatment with agent X was considered with a division of the total receptor concentration ([R_0_]) by ≈5.556 (Table 3), simulating that 18% of the A_1_ adenosine receptors remained intact (in agreement with our previous results [10]). In Model 4 and its variants, if a transport inhibitor was present, the agent X pretreatment was considered with a second procedure as well, by introducing a further c_bias_ value (see: next paragraph and Table 3). Moreover, in Model 4-v2 (one of the variants), if a transport inhibitor was present, the agent X pretreatment was considered with a third procedure as well, through an additional division by 14.8952, when computing the “near-receptor” concentration of agonist A (see Section 4.6).

NBTI, a nucleoside transport inhibitor, was modelled by an agent NB. Its effect was taken into account by omitting the division of the exogenous agonist A concentration “in the organ bath” by 400, or by dividing it using a smaller number (as anticipated above,), and, in addition, by considering a surplus endogenous agonist A concentration (c_bias_), when computing an effect. The earlier models contained only one c_bias_ (Models 1–3), whereas the final one (Model 4) and its two variants possessed two c_bias_ values: one for a mere NB treatment, and the other one for an X + NB co-treatment. In Models 1 and 2, c_bias_ was an arbitrary value, while in Models 3 and 4, c_bias_ values equaled the surplus interstitial adenosine concentrations determined in the most recent ex vivo study [14] (Table 3). As mentioned at the end of the previous paragraph, in Model 4 and its variants, an interaction between agent X pretreatment and agent NB treatment was also taken into account.

Effect values of the simulated E/c curves were plotted against the “bathing medium concentrations” of the given exogenous agonist, as usually these concentrations are only known during the in vivo and ex vivo experiments.

### 4.3. Construction of Simple E/c Curves (Expressing the Action of One Agonist in a Standard Way)

Effect values evoked by one agonist (C or A) were determined in terms of the operational model of agonism using the following equation (equivalent to Equation (10) in [21]):(1)$$E=E_{m}\times \frac{{\left[R_{0}\right]}^{n_{op}}\times c^{n_{op}}}{K_{E}{}^{n_{op}}\times {\left(K+c\right)}^{n_{op}}+{\left[R_{0}\right]}^{n_{op}}\times c^{n_{op}}}$$
where E: the effect value; E_m_: the upper asymptote of the signal amplification function of the system; [R_0_]: the total concentration of the operable receptors (defining a naïve receptor population and a reduced one); c: the “near-receptor” concentration of the (exogenous) agonist C or A (computed from the agonist concentration “in the organ bath”); K: the equilibrium dissociation constant of the agonist-receptor complex (an inverse measure of agonist affinity); K_E_: an inverse measure of agonist efficacy; n_op_: the operational slope factor.

The so-called “organ bath” concentrations of agonists C and A ranged from 10^−10^ to 10^−2.5^ (~3.1623∙10^−3^) mol/L. The “near-receptor” concentrations (c values for Equation (1)) were calculated according to the nature of the simulated agonist and treatment. To ensure the best comparability to Figure 1 (showing ex vivo results from [14]), concentrations depicted in Figure 2, Figure 3 and Figure 4 for agonists C and A ranged between 10^−10^ to 10^−4^ and 10^−9^ to 10^−2.5^ mol/L, respectively.

Atria without and with an FSCPX pretreatment were simulated using [R_0_] equaling 10^−10^ and 1.8·10^−11^, respectively. These settings agree with our previous ex vivo result, i.e., approximately 10–20% of the original A_1_ adenosine receptors remained operable after a 45-min-long pretreatment with 10 µmol/L FSCPX (followed by a 75-min washout period) in the guinea pig atrium [10]. Effect values calculated with Equation (1) were plotted against the organ bath concentrations of the given agonist.

### 4.4. Construction of Complex E/c Curves (Representing the Co-Action of Two Agonists with the Neglect of One of Them)

Effect values elicited by a surplus concentration of the endogenous agonist A produced by agent NB (c_bias_) together with increasing concentrations of the exogenous agonist C or A (c_test_) were computed in two steps. First, intermediate effect values were generated by means of the following equation (equivalent with Equation (7) in [28]):(2)$$E=E_{m}\times \frac{{\left(\tau_{\mathrm{test}}\times c_{\mathrm{test}}\times K_{\mathrm{bias}}+\tau_{\mathrm{bias}}\times c_{\mathrm{bias}}\times K_{\mathrm{test}}\right)}^{n_{\mathrm{op}}}}{{\left(c_{\mathrm{test}}\times K_{\mathrm{bias}}+K_{\mathrm{test}}\times K_{\mathrm{bias}}+c_{\mathrm{bias}}\times K_{\mathrm{test}}\right)}^{n_{\mathrm{op}}}+{\left(\tau_{\mathrm{test}}\times c_{\mathrm{test}}\times K_{\mathrm{bias}}+\tau_{\mathrm{bias}}\times c_{\mathrm{bias}}\times K_{\mathrm{test}}\right)}^{n_{\mathrm{op}}}}$$
where E: the intermediate (still “unbiased”) effect; E_m_: the upper asymptote of the signal amplification function of the system; c_bias_: the surplus (near-receptor) concentration of the endogenous agonist A; c_test_: the near-receptor concentration of the exogenous agonist C or A (computed from the agonist concentration in the organ bath); K_bias_ and K_test_: K values (see: Equation (1)) for the agonist providing c_bias_ and c_test_, respectively; τ_bias_ and τ_test_: [R_0_]/K_Ebias_ and [R_0_]/K_Etest_, respectively, where K_Ebias_ and K_Etest_ are K_E_ values (see: Equation (1)) for the agonist supplying c_bias_ and c_test_, respectively.

Subsequently, the intermediate effect values were transformed (“biased”) using the following relationship (equivalent to Equation (5) in [19], and identical with Equation (5) in [20]):(3)$$E^{\prime }=100-\frac{100\times \left(100-E\right)}{100-E_{bias}}$$
where E′: the final (“biased”) effect value; E: the intermediate effect value provided by Equation (2); E_bias_: the effect elicited by c_bias_ (computed with Equation (2) by setting c_test_ at zero, with other parameters being the same as for the corresponding E).

The biasing transformation by means of Equation (3) simulated the neglect of c_bias_ and its effect (E_bias_) during the evaluation. Upon in vivo and ex vivo experiments, of course, an E/c curve, which is generated in the presence of a c_bias_, would possess unbiased effect values and reflect the relationship between the c_bias_+c_test_ concentrations and the unbiased effect values (as indicated by Equation (2)), if it were evaluated properly (tailored to the situation and not a standard way). However, c_bias_ and E_bias_ are typically unknown (or ignored). Final effect values yielded by Equation (3) were plotted versus the organ bath concentrations of the agonist providing c_test_.

### 4.5. Characterization of the Simulated E/c Curves

All E/c curves were fitted to the Hill equation (used in the following form that is identical to Equation (10) in [31]):(4)$$E=\frac{E_{max}}{1+10^{n\times \left(logEC_{50}-logc\right)}}$$
where E: the effect; E_max_: the maximal effect achievable with the given agonist in the given system; EC_50_: the agonist concentration in the organ bath that produces half-maximal effect; n: the Hill slope factor; c: the agonist concentration in the organ bath.

Use of the bathing medium (rather than near-receptor) concentrations for Equation (4) agrees with the curve fitting practice of in vivo and ex vivo experiments (where near-receptor concentrations are usually unknown).

### 4.6. Evolution of the In Silico Models

The present in silico investigation started from the initial model of our recent computer simulation study [13]. The model parameters were being changed to achieve the greatest similarity to eight E/c curves that were selected from our recent ex vivo investigation [14] (and are presented herein in Figure 1). This procedure yielded four models (Models 1–4, see Table 3) and two variants of the last model (Models 4-v1 and 4-v2).

In all models, agonists C and A shared the same properties, except for the inward transmembranous transport, which was characteristic only to agonist A.

In Model 1, the simplest model exhibiting some similarities with the original ex vivo E/c curves to be modelled [14], agent NB was supposed to totally stop the transport of agonist A (i.e., division of the organ bath concentration of agonist A by 400 was omitted to receive the near-receptor concentration during the calculation of effect values for the NB-treated and X + NB co-treated E/c curves of agonist A). The endogenous and exogenous adenosine were handled totally separately (for all models, excepting Model 4-v2, see below). In Model 1, only one (arbitrary) c_bias_ value was used, so no interaction was assumed between the effects of agent X and agent NB (Table 3).

Model 2 stemmed from Model 1 through one modification, i.e., E_m_ value was decreased from 100 to 99 (Table 3).

Model 3 derived from Model 1 (or, equivalently, from Model 2) via three modifications. First, E_m_ was set to 90. Second, to consider the effect of agent NB (solely or along with agent X), c_bias_ was fixed at a constant value, namely c_x_ in the case of the sole NBTI treatment from [14] (i.e., the CPA concentration equieffective with the surplus interstitial concentration of endogenous adenosine produced by NBTI treatment without a previous FSCPX pretreatment). With the use of only one c_bias_, no interaction was considered between effects of agents X and NB (Table 3). Third, in Model 3, it was hypothesized that agent NB only slowed (and not stopped) the agonist A transport. Accordingly, the organ bath concentration of agonist A was divided by 6 to get the near-receptor concentration, when computing effect values for the NB-treated and X + NB co-treated E/c curves of agonist A.

Into Model 4, both c_x_ values of [14] were introduced as c_bias_ values, i.e., CPA concentrations equieffective with the extra interstitial concentration of endogenous adenosine caused by NBTI without and with an FSCPX pretreatment. The newly introduced c_bias_ was used to define the surplus endogenous agonist A concentration for the X + NB co-treated E/c curves. Use of two c_bias_ values implied an interaction between the effects of agents X and NB (Table 3).

Model 4-v1, a variant of Model 4, differed from Model 4 in that no agonist A transport was supposed in it under the effect of agent NB (just like in Models 1 and 2). So, the effect values of the NB-treated as well as X + NB co-treated E/c curves were computed from the organ bath concentrations of agonist A (of course, the NB treatment and X + NB co-treatment were distinguished by using different c_bias_ values, as described for Model 4).

In Model 4-v2, the other variant of Model 4, agent X was supposed to influence the action of agent NB on the level of the exogenous agonist A as well (and not only on the level of the endogenous one). This unique feature of Model 4-v2, the only difference between Models 4-v2 and Model 4, exhibited a previous hypothesis that emerged in [13], i.e., FSCPX exerts its effect via modifying ENT1, the target of NBTI. The extent of the influence of agent X on the concentration of exogenous agonist A was defined (somewhat arbitrarily) as follows: when calculating the effect values, the near-receptor concentrations for the X + NB co-treated E/c curve of agonist A were computed through dividing the corresponding near-receptor concentrations for the solely NB treated E/c curve of agonist A by 14.8952. This value is the ratio of the two c_x_ values from [14] (c_x_ measured during the mere NBTI treatment per c_x_ determined during the FSCPX + NBTI co-treatment).

### 4.7. Computer Simulation and Data Analysis

The agonist concentration (c) and EC_50_ were expressed as common logarithms in Equation (4), because Equation (4) was used for curve fitting. Effect values of Equations (1), (2) and (3) as well as E_bias_ in Equation (3) were computed with Microsoft Excel 2016 (Microsoft Co., Redmond, WA, USA). For curve plotting and fitting, GraphPad Prism 8.1.1 for Windows (GraphPad Software Inc., La Jolla, CA, USA) was used.

## 5. Conclusions

Our goal was to perform an in silico investigation regarding the background of a paradoxical phenomenon, first described in [5], sc. the irreversible A_1_ adenosine receptor antagonist FSCPX apparently increases the maximal response to adenosine in the presence of NBTI. The present study used an improved approach for computer simulation (modelling the adenosine homeostasis and A_1_ adenosinergic control of contractility in the isolated, paced guinea pig left atrium) by addressing the issue of E_m_ parameter of the operational model of agonism. Taking outcomes of the current in silico study together with our previous in silico and ex vivo results, we have concluded that the above-mentioned paradoxical phenomenon can be ascribed to two independent, simultaneous and additive factors. One is the interesting phenomenon that forms the basis of RRM [19], i.e., interstitial accumulation of the endogenous and exogenous adenosine exerts the opposite effect on the E/c curve of adenosine in our experimental arrangement (as described in [5]). The other factor underlying the paradoxical phenomenon investigated herein is an interference between effects of two adenosine analogues, FSCPX and NBTI, in our experimental setting that was first suggested in [13] and was first evidenced (ex vivo) in [14]. Herein, we have provided in silico evidence for this interference, proposing that FSCPX, in addition to antagonizing the A_1_ adenosine receptor, blunts the interstitial accumulation of endogenous (but not exogenous) adenosine produced by NBTI.

## Figures and Tables

**Figure 1 molecules-24-02207-f001:**
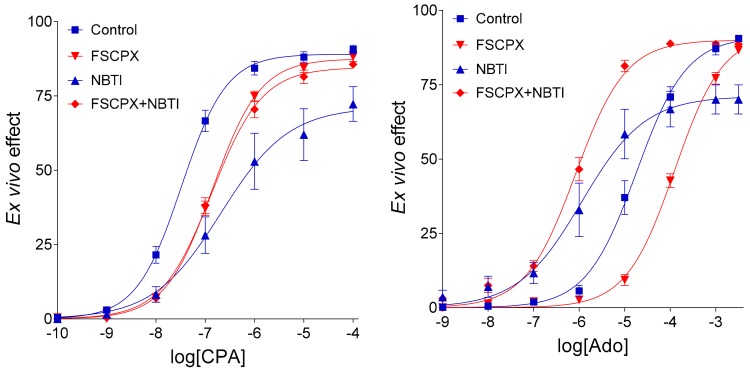
Concentration-response (E/c) curves of CPA, a synthetic full agonist of the A_1_ adenosine receptor (having relatively long half-life, see: [15]), and adenosine, the physiological adenosine receptor full agonist (possessing very short half-life, see: [15]), where the direct negative inotropic response of isolated, paced guinea pig left atria were measured. The E/c curves illustrate the influence of NBTI, a nucleoside transport inhibitor, on the effect of CPA and adenosine, without (blue curves) and with (red curves) a pretreatment with FSCPX, a chemical known as an irreversible A_1_ adenosine receptor antagonist. The *x*-axis shows the common logarithm of the molar concentration of the given agonist (in the bathing medium), while the *y*-axis denotes the direct negative inotropic effect (determined as a percentage decrease of the initial contractile force). The symbols show the averaged responses (± SEM), and the lines represent the fitted Hill equation. CPA: *N*^6^-cyclopentyladenosine; Ado: adenosine; NBTI: *S*-(2-hydroxy-5-nitrobenzyl)-6-thioinosine; FSCPX: 8-cyclopentyl-*N*^3^-[3-(4-(fluorosulfonyl)benzoyloxy)propyl]-*N*^1^-propylxanthine. Data are redrawn from [14].

**Figure 2 molecules-24-02207-f002:**
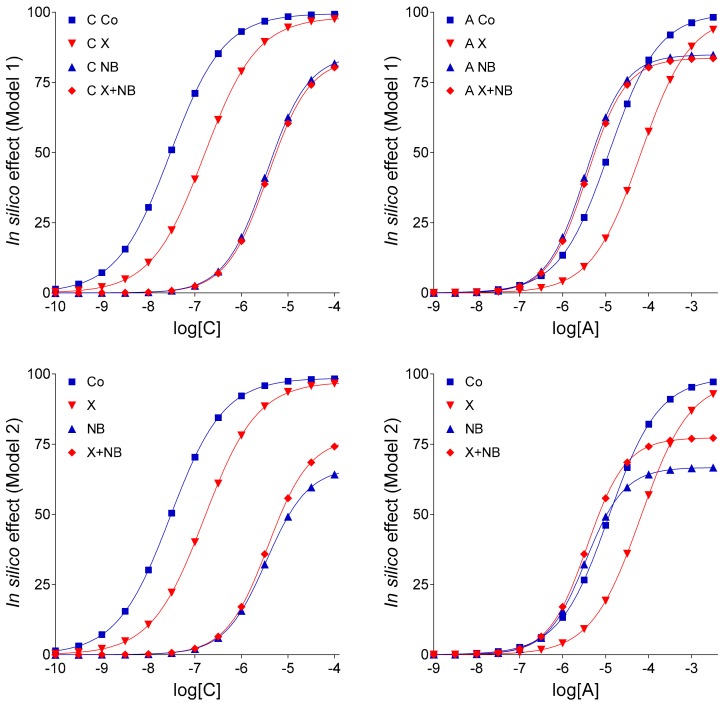
Simulated concentration-response (E/c) curves of agonist C (representing CPA, a synthetic A_1_ adenosine receptor agonist with long half-life) and agonist A (symbolizing adenosine, the physiological adenosine receptor agonist with short half-life), according to Models 1 and 2 (modelling the supraventricular myocardium, where A_1_ adenosine receptor agonists exert strong direct negative inotropic effect). The only difference between Model 1 and Model 2 is the value of the E_m_ parameter (100 and 99, respectively, see Equations (1) and (2)). The E/c curves show the influence of agent NB (representing NBTI, a nucleoside transport inhibitor) on the effect evoked by agonist C and A, without (blue curves) and with (red curves) a pretreatment with agent X (representative of FSCPX, a chemical known as an irreversible A_1_ adenosine receptor antagonist). The *x*-axis denotes the common logarithm of the molar concentration of the given agonist (in the organ bath), and the *y*-axis shows the effect. The lines represent the fitted Hill equation (see Equation (4)). CPA: *N*^6^-cyclopentyladenosine; FSCPX: 8-cyclopentyl-*N*^3^-[3-(4-(fluorosulfonyl)benzoyloxy)propyl]-*N*^1^-propylxanthine; NBTI: *S*-(2-hydroxy-5-nitrobenzyl)-6-thioinosine.

**Figure 3 molecules-24-02207-f003:**
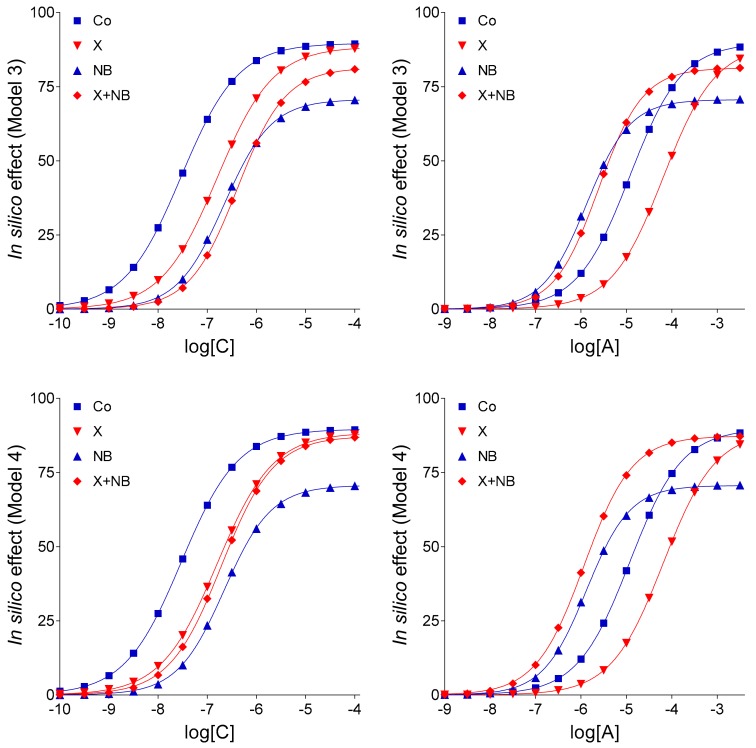
Simulated concentration-response (E/c) curves of agonist C (representing CPA, a synthetic A_1_ adenosine receptor agonist with long half-life) and agonist A (symbolizing adenosine, the physiological adenosine receptor agonist with short half-life), according to Models 3 and 4 (modelling the supraventricular myocardium, where A_1_ adenosine receptor agonists exert strong direct negative inotropic effect). The E/c curves show the influence of agent NB (representing NBTI, a nucleoside transport inhibitor) on the effect evoked by agonist C and A, without (blue curves) and with (red curves) a pretreatment with agent X (representative of FSCPX, a chemical known as an irreversible A_1_ adenosine receptor antagonist). The *x*-axis denotes the common logarithm of the molar concentration of the given agonist (in the organ bath), and the *y*-axis indicates the effect. The lines represent the fitted Hill equation (see: Equation (4)). CPA: *N*^6^-cyclopentyladenosine; FSCPX: 8-cyclopentyl-*N*^3^-[3-(4-(fluorosulfonyl)benzoyloxy)propyl]-*N*^1^-propylxanthine; NBTI: *S*-(2-hydroxy-5-nitrobenzyl)-6-thioinosine.

**Figure 4 molecules-24-02207-f004:**
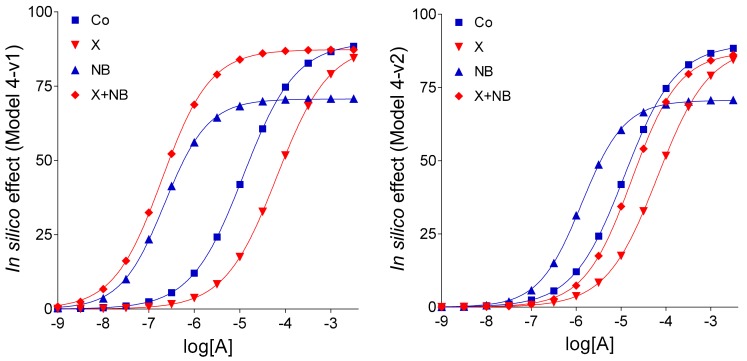
Simulated concentration-response (E/c) curves of agonist A (representing adenosine, the physiological adenosine receptor agonist with short half-life), according to Models 4-v1 and 4-v2 (modelling the atrial myocardium, where A_1_ adenosine receptor agonists evoke strong direct negative inotropy). The E/c curves denote the influence of agent NB (modelling NBTI, a nucleoside transport inhibitor) on the effect of agonist A, without (blue curves) and with (red curves) a pretreatment with agent X (representing FSCPX, a chemical known as an irreversible A_1_ adenosine receptor antagonist). The *x*-axis shows the common logarithm of the molar concentration of agonist A (in the organ bath), and the *y*-axis denotes the effect. The lines represent the fitted Hill equation (see: Equation (4)). FSCPX: 8-cyclopentyl-*N*^3^-[3-(4-(fluorosulfonyl)benzoyloxy)propyl]-*N*^1^-propylxanthine; NBTI: *S*-(2-hydroxy-5-nitrobenzyl)-6-thioinosine.

**Table 1 molecules-24-02207-t001:** The Hill parameters of the CPA and adenosine concentration-response (E/c) curves of Figure 1.

**CPA**	**Control**	**FSCPX**	**NBTI**	**FSCPX + NBTI**
E_max_	89.21 ± 1.46	87.49 ± 1.75	71.68 ± 5.24	84.6 ± 1.64
logEC_50_	−7.47 ± 0.074	−6.85 ± 0.05	−6.53 ± 0.32	−6.86 ± 0.06
*n*	0.98 ± 0.03	0.93 ± 0.04	0.84 ± 0.09	0.86 ± 0.02
**Ado**	**Control**	**FSCPX**	**NBTI**	**FSCPX + NBTI**
E_max_	91.04 ± 1	93.85 ± 1.78	71.23 ± 4.9	90.22 ± 1
logEC_50_	−4.74 ± 0.13	−3.88 ± 0.07	−5.92 ± 0.25	−6.08 ± 0.07
*n*	0.85 ± 0.03	0.83 ± 0.07	0.81 ± 0.13	0.83 ± 0.1

E_max_, logEC_50_ and n (mean ± SEM) are best-fit values of the Hill equation fitted to the individual E/c curves. SEM: standard error of the mean (for more detail, see: Figure 1). Data are from [14].

**Table 2 molecules-24-02207-t002:** The Hill parameters of the concentration-response (E/c) curves that were simulated according to the Model 4 and are presented in Figure 3.

**C**	**Co**	**X**	**NB**	**X + NB**
E_max_	89.61 ± 0.03	88.45 ± 0.07	70.67 ± 0.12	87.27 ± 0.12
logEC_50_	−7.53 ± 0.001	−6.8 ± 0.003	−6.66 ± 0.005	−6.71 ± 0.005
*n*	0.75 ± 0.001	0.76 ± 0.003	0.91 ± 0.008	0.82 ± 0.006
**A**	**Co**	**X**	**NB**	**X + NB**
E_max_	89.76 ± 0.01	89 ± 0.04	70.61 ± 0.14	87.27 ± 0.15
logEC_50_	−4.92 ± 0.0003	−4.19 ± 0.0008	−5.88 ± 0.005	−5.93 ± 0.005
*n*	0.75 ± 0.0004	0.75 ± 0.0007	0.91 ± 0.009	0.82 ± 0.006

E_max_, logEC_50_ and n (mean ± SE) are best-fit values of the Hill equation (Equation (4)) fitted to the simulated E/c curves. SE (standard error) characterizes how precisely the given best-fit value has been determined (for more detail, see: Figure 3).

**Table 3 molecules-24-02207-t003:** Parameters of the operational model of agonism defining four in silico models.

	Model 1	Model 2	Model 3	Model 4
**E_m_**	100	99	90	
**n_op_**	75			
**[R_0_]**	10^−10^ (Co) 1.8 × 10^−11^ (X)			
**K**	3 × 10^−5^			
**K_E_**	10^−13^			
**c_bias_**	2.5 × 10^−6^ (NB and X + NB)		1.002 × 10^−7^ (NB and X + NB)	1.002 × 10^−7^ (NB) 6.727 × 10^−9^ (X + NB)

Co: no simulated (pre)treatment; X: simulated pretreatment with agent X; NB: simulated treatment with agent NB; X + NB: simulated pretreatment with agent X followed by a treatment with agent NB (briefly, X + NB co-treatment). For an explanation of abbreviations in the first column, see Equation (1) and (2) below.

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
