# Peer review of "An Advanced in Silico Modelling of the Interaction between FSCPX, an Irreversible A1 Adenosine Receptor Antagonist, and NBTI, a Nucleoside Transport Inhibitor, in the Guinea Pig Atrium"

_molecules, 2019, doi:10.3390/molecules24122207_

Round 1

Reviewer 1 Report

The authors have applied a mathematical model of the adenosine handling mechanisms and A1 adenosinergic control of contractility of the isolated, paced guinea pig left atrium, to reproduce the E/c curves. The manuscript is potentially of interest, however need some modifications.

Abstract:

-the abstract is circumvented and the authors should rewritten the abstract simplified the way to expose the experimental design and the data.

-“ex vivo” is a  latin word and needs to be written in Italic in all the manuscript (like “in vivo”, off course).

Introduction:

-        The caption of figures:” Figure 1. Concentration-response (E/c) curves of CPA, a synthetic full agonist of the A1 adenosine receptor with relatively long half-life [15], and adenosine, the physiological adenosine receptor full agonist with very short half-life [15], where the direct negative inotropic response of isolated, paced guinea pig left atria were measured. The E/c curves illustrate the influence of NBTI, a nucleoside transport inhibitor, on the effect of CPA and adenosine, without (blue curves) and with (red curves) a pretreatment with FSCPX, a chemical known as an irreversible A1 adenosine receptor antagonist. The x-axis shows the common logarithm of the molar concentration of the given agonist (in the bathing medium), while the y-axis denotes the direct negative inotropic effect (determined as a percentage decrease of the initial contractile force). The symbols show the averaged responses (±SEM), and the lines represent the fitted Hill equation. .....Data are redrawn from [14].”

1.      Question on figure 1: The data are redrawn from the previous paper 14 or 15? It is not clear. 

2.      Moreover, is it correct shows in figure 1 and table 1 the already published data in this manuscript? The authors should put this figure/legend and the table 1  in supplementary data section and exclude from the manuscrit.

-        The sentences: “In contrast, exogenous adenosine is accumulated by NBTI during the construction of the E/c curve, so it can elicit a greater effect. This distinction between effects of the endogenous and exogenous adenosine in the presented experimental arrangement forms the basis for the so-called receptorial responsiveness method (RRM) [19,20], theoretical concept of which was essential for thecurrent work.

1.      ” Why “during” is in Italic?

2.      The second sentence must be re-written in English

Results: The authors should to focus the data on model 3 and 4 that present more innovation as compare to model 1 and 2

Discussion: “Putting this earlier result into a biological context, this may indicate that FSCPX modifies ENT1 in a way that the altered ENT1 can carry adenosine well, but it can be less inhibited by NBTI. In the present work, however, the paradoxical phenomenon in question could be reproduced in an alternative way, unamely by decreasing the Em parameter of the operational model of agonism (as shown by Models 1 and 2 in Fig. 2).”

-The authors have discussed the data considering only their previous their works, but not of other models concern in ex vivo and in vivo models in vertebrates:

·   

·   

·        Progress in Biophysics and Molecular Biology 116 (1), (2014), Pages 40-47

·        Exp Cell Res. 2018 Aug 1;369(1):166-175. doi: 10.1016/j.yexcr.2018.05.022

            Cell Death Dis. 2018 Jun; 9(6): 650. doi: 10.1038/s41419-018-0609-7

·      

Author Response

Response to Reviewer 1

The authors are grateful for your work to review our manuscript. All your concerns and recommendations have been addressed. Our answers to the comments are as follows (in sequence):

Abstract:

-        We made effort to clarify the structure of the Abstract.

-        We agree with the Reviewer and generally use Italics for certain Latin phrases. Here, in the Molecules, the journal’s style makes us avoid Italics in the text.

Introduction:

-        We have made obvious which reference applies to data plotted in Fig. 1.

-        We feel that it is acceptable to recapitulate data previously published by the same team in the same journal if they are important for the new results. However, we have asked the Editor’s position for this.

-        We wanted to highlight this word. Nevertheless, consistent with the point 2 above, the format bold and Italics has been changed to bold and underlined.

-        We have rewritten the sentence in question.

Results:

-        We have completed the description of Model 4, as suggested.

Discussion:

-        To briefly introduce our own results, works of others dealing with models concerning the heart have been acknowledged at the beginning of the Discussion.

Overall, the manuscript was reworked to improve its clarity and informativeness. Thank You again for reviewing our work. We hope that you will find our revised manuscript suitable for publication in the Molecules.

Reviewer 2 Report

The authors of this manuscript propose several mathematical models of the E/C curves which may reproduce the behaviors of FSCPX under the several conditions, and a model among them (model 4) can reproduce the experimental data. I think that the results in this manuscript are very interesting and useful. However, I also feel that the following points are reconsidered before publication.

1. The authors describe the details of the several models in the section “Materials and Methods”. Some of them have been already discussed in their previous papers. However, for readers the action process of the proteins and chemical compounds in the system should be presented and the derivation of a model (ex. Equation 1) should be simply explained in this section.

2. In general, a mathematical model with several parameters can sometimes reproduce a result (functional shape), which is convenient for authors, by just simple tuning of some parameters. The authors should give explanations that the present model (model 4) really reproduces the experimental data, for example with robustness of the results in site of the changes of the values of parameters.

3. The present results should be examined by the experiments of molecular level. The authors should give some possibility on this in the discussion section.

Author Response

Response to Reviewer 2

Thank You for reviewing our manuscript. All concerns and suggestions have been addressed. The answers to the Reviewer’s comments are as follows (in sequence):

1.        The Materials and Methods section has been extended by inserting a new subsection, in which the biological background of this in silico work was summarized.

2.        The authors agree with the Reviewer that mathematical models can be misleading. In this study, we have investigated whether certain assumptions can be harmonized with biological data, rather than statistically compare our models with one another. The robustness of outcomes was not determined either. Nevertheless, to picture the reliability of Model 4 (that might be missing from the original manuscript), we stated in the Results that moderate changes in the values of parameters for Model 4 did not cause dramatic changes in the E/c curves, contrary to that was experienced in the case of Em close to 100 in Models 1 and 2.

3.        We provided a potentially useful direction for further investigations in the penultimate paragraph of the Discussion.

Overall, the manuscript underwent an extensive revision to improve its clarity and informativeness. Thank You again for reviewing our work. We hope that you will find the present version of our manuscript suitable for publication in the Molecules.

Round 2

Reviewer 1 Report

The authors have improved the paper, anyway some splee check is even need because some words are attach the one to the others.